# The Side-Release Method Measures the High-Pressure Sound Velocity of Iron Using Line-Spatially Resolved DISAR

**DOI:** 10.3390/mi15091165

**Published:** 2024-09-20

**Authors:** Long Chen, Cangli Liu, Longhuang Tang, Heli Ma, Xing Jia, Tianjiong Tao, Shenggang Liu, Yongchao Chen, Xiang Wang, Jian Wu, Chengjun Li, Dameng Liu, Jidong Weng, Huan Liu

**Affiliations:** 1National Key Laboratory of Shock Wave and Detonation Physics, Institute of Fluid Physics, China Academy of Engineering Physics, Mianyang 621900, China; chenlongcaep@163.com (L.C.); cangliliu@aliyun.com (C.L.); tanglonghuang@tju.edu.cn (L.T.); marcos12@126.com (H.M.); jiaxing@caep.cn (X.J.); zjuttj@163.com (T.T.); liushenggangpla@126.com (S.L.); ycchen16@fudan.edu.cn (Y.C.); xiangwang102@126.com (X.W.); ceuwj@zju.edu.cn (J.W.); lstrus@126.com (C.L.); 2State Key Laboratory of Tribology in Advanced Equipment, Department of Mechanical Engineering, Tsinghua University, Beijing 100084, China; ldm@tsinghua.edu.cn

**Keywords:** shock loading, surface structure, sound velocity, laser velocimetry

## Abstract

The study of high-pressure sound velocity is an important part of shock wave physics, and the study of ultra-high pressure sound velocity of iron is of great significance to many research fields such as geophysics, solid state physics, and crystallography. At present, the measurement of sound velocity is usually carried out by the catch-up sparse wave method and windowed VISAR technology, which is complex in structure and not highly adaptable. In particular, for the ultra-high pressure sonic velocity measurement of metals, it is limited by the loading platform and window materials and cannot realize the high temperature and high-pressure environment of the earth’s inner core. In this paper, the sound velocity measurement of iron under high temperature and high-pressure environment (78 GPa) is realized based on the two-stage light gas cannon experimental platform. The side-side sparse wave method was used to establish a coupling model of high-spatially resolved optical group and fiber bundle. A multiplexed all-fiber laser interferometry velocity measurement system (DISAR) was built, and the spatial resolution was better than 20 μm. In this paper, we will provide a feasible route for a method for measuring the high spatiotemporal resolution velocity.

## 1. Introduction

The Earth’s core is the most important spherical structure in the Earth’s interior, and its internal pressure-temperature state varies from 136 GPa and about 4000 K to 364 GPa and 5000~6000 K. Preliminary Reference Earth Model (PREM) studies have shown that the Earth’s core is composed of two parts: a liquid outer core and a solid inner core. As we known, iron is the main element in the earth’s core, and the study of iron’s equation of state is of great significance to many research fields such as geophysics, solid state physics, and crystallography [1]. Sound velocity is the basic property of matter under high pressure, and high-pressure sound velocity is one of the important parameters to characterize the mechanical properties of materials under high pressure, and it is also an important parameter to judge whether a solid material undergoes phase transformation under high temperature and high pressure [2]. The structural state of the surface of the material will also affect its high-pressure physical properties, which is also a direction of sound velocity research. Accurate measurement of the sound velocity of matter under high pressure can be used to infer the internal structure and material composition of celestial bodies such as planets, and the sound velocity at extreme high pressure is also an important part of the concerns of materials science, condensed matter physics, and (dense) plasma physics, especially when the quantum effect above TPa pressure plays a decisive role in the atomic structure, the sound velocity measurement results can also verify the molecular dynamics model. By measuring the high-pressure sound velocity of iron, it is possible to determine its physical and mechanical parameters such as yield strength, shear modulus, Gruneisen coefficient, and the equation of state of iron, which is of great significance for the study of the composition and properties of matter in the high-temperature and high-pressure environment of the earth’s core.

The study of the sound velocity of iron began in 1886 [3], and by 1960 Al’tshuler et al. proposed a method for measuring the velocity propagation of small perturbations after shock waves [4]. The relationship between sound velocity and density of iron and its compounds at different pressures is in accordance with the “Birch’s Law” results [5]. However, the state of earth’s inner core is in accordance with 360~500 GPa and 5000~6000 K, which is also the preamble that geophysics is most concerned about. It is difficult to measure the sound velocity of iron by realizing the high-pressure and high-temperature state of the earth’s core on the two-stage light gas gun [6]. The side-release method can be used to measure higher pressure sound velocities in the three-stage light gas gun.

Typically, the high-pressure sound velocity of a material is measured using the sparse wave catch-up method. In 1960, Al’tshuler et al. first used the “chasing sparse wave” method to measure the speed of sound after a shock wave in a metal [4]. However, the “chasing sparse wave” method measures the sound velocity after the shock wave of a metal mainly using VISAR (Velocity Interferometer for any Reflector) [7] and optical analysis (Optical Analysis) techniques. In both methods, the unloading sound velocity of the material to be measured is calculated by the time course of the unloading sparse wave at the recorded flyer interface chasing the propagating shock wave in the sample from the time course of a thin flyer striking the sample [8]. In the photo analysis technique, multi-step samples are used to improve the fitting accuracy of the catch-up thickness, the structure of the device is complex, and the test accuracy is greatly affected by the interface wave system. At the same time, the sample size is limited by the experimental loading platform. The windowed VISAR technology can obtain both particle velocity and sound velocity history, but it has high requirements for the optical properties of the window material [9,10]. There are not many materials that can be used as a window at present, and only a few metal samples can find impedance matching window materials, such as monocrystalline lithium fluoride and sapphire, whose impedances are similar to those of aluminum and steel, respectively. The bigger problem is that the window material becomes opaque at ultra-high pressures, causing the laser to fail to penetrate and leading to measurement failures. Therefore, in the actual measurement, the step sample cannot be enlarged due to the loading platform, and the window material is limited by impedance matching and impact pressure, which makes it difficult to accurately and directly measure the ultra-high pressure sound velocity of the metal by using the “catching up with sparse wave” method.

Another method of “side sparse wave” also measures the speed of sound after a shock wave of a metal. In 1960, Al’tshuler and Kormer et al. measured the longitudinal wave sound velocity of metallic materials such as aluminum, copper, lead, and iron at pressures up to 100 GPa using the effect of marginal sparse waves on the plane range of a one-dimensional shock wave in a sample and the effect of interfacial sparse waves from a flyer on the shock wave velocity in a sample [4]. When the high-speed flyer hits the target, the sample is in the high-pressure state of the shock wave, but the side positions of the flyer, the target plate, and the sample to be measured are all in the state of zero pressure, and the sparse wave from the side part of the experimental device will interfere with the propagation and post-wave state of the plane shock wave. The wave head of the lateral sparse wave propagates at the speed of sound C with respect to the matter behind the shock wave. By definition, the speed of sound is the speed at which a small perturbation wave propagates with respect to the wavefront medium. Since the wavefront velocity is C + U > D (U is the bulk velocity of the material after the wavefront), any initial perturbation catches up with the wavefront, producing variations in all parameters, including the kinetic parameters U and D. Since it indicates that the location of the acoustic disturbance is at the wavefront, this allows the measurement of the speed of sound [8].

Thus, the velocity of sound can be obtained by measuring the sparsity angle, if the D and U of the shock wave are known, and the tangent of the sparsity angle needs to be measured experimentally. However, due to the presence of elastic longitudinal waves, it is difficult to accurately determine the boundaries of the unloading zone using optical photographs, and the resulting sound velocity measurement is not accurate. Then, in order to accurately measure the position of point B, a high-spatially resolved linear/area array close-row velocimeter probe is required [11].

In 2016, Malone developed a PDV (Photon Doppler Velocimetry) microlens probe that can measure the velocity information of 37 different points in a field of view less than 1 mm, and analyze the change behavior of grain boundaries under shock loading [12], with a spatial resolution of up to 37 μm. Measuring the changing behavior of microstructures under shock loading requires the development of small-field close-packed surface array fiber optic probes. In 2018, Li Mu et al. [13] found that the boundary between the lateral sparse region and the plane impact region can be directly obtained by the line VISAR, so a more stringent side sparsity condition was designed to ensure that the lateral sparse wave intersecting with the shock wave front was introduced when the shock wave entered the sample, and the bulk sound velocity on the main impact insulation line of the quartz single crystal was continuously measured, with a pressure range from 0.25 to 1.45 Tpa.

Displacement interferometers (displacement interferometer system for any reflector, DISAR) [14] have simple structure and wide applicability due to the advantages of optical fiber communication technology, but they are limited by the small numerical aperture of single-mode optical fibers, and the array close-row mode cannot fundamentally solve the problem of high spatial resolution of the diffraction limit bottleneck, especially in one-dimensional/two-dimensional space measurement, it cannot have the dual advantages of measuring depth of field length and high spatial resolution [15].

In this paper, we will study the methods and testing techniques that can be used to measure the ultra-high pressure sound velocity of metals. The first is the optimal design of the optical system of the high-spatially resolved linear array/area array close-packed fiber probe, the second is the research on multi-channel laser Doppler velocimetry technology based on wavelength division multiplexing technology, and the third is based on the experimental method of ultra-high pressure sound velocity of metals (such as iron, aluminum, etc.) by the “side sparse wave” method. It can not only improve the ability of laser interferometry with high spatial and temporal resolution, but also provide a feasible route for the experimental method of ultra-high pressure sound velocity. We firstly take advantage of the side-release method to avoid the problem of the catch-up method being unmeasurable under higher pressure because the window material is opaque. Secondly, the use of line-spatially resolved fiber optic probe can be applied to smaller targets to measure sound velocity at higher pressures on some impact-loading platforms where target size is limited. The combination of the two advantages can obtain the sound velocity in the earth’s inner core under shock loading. The technical route for measuring high-pressure sound velocity proposed in this paper can be better applied to the three-stage light gas gun or magnetic drive loading platform that is not limited by the target size and window material.

## 2. Principle

### 2.1. Side Sparse Method

As shown in the Figure 1, the lateral part of the “shoulder” of the device serves as the source of sparse waves [16]. The expansion wave propagates inside the metal at the speed of sound and catches up with the wavefront, causing a drop in pressure in the edge region of the sample. Consider the two positions of moment t and *t +* Δ*t*. Point O distinguishes between sparse and unsparse boundaries. After Δ*t*, the shock wave travels at a distance *D·*Δ*t* and the metal particles initially at the point O move *U·*Δ*t*. At the same time interval Δ*t*, the sparse wave propagates with a sphere of radius *C·*Δ*t* to the point O_1_. Due to the sparse wave motion, the shock wave propagation forms an angle, and the trajectory of the boundary is a straight line, forming a constant sparse angle α along the direction of the shock wave propagation. In the right triangle O_1_AB, there is $AB=\sqrt{{\left(C∆t\right)}^{2}-{\left(D-U\right)}^{2}∆t^{2}}$. At the same time in the triangle AOB, AB = *D·*Δ*t·tanα*. Thus, we can obtain the expressions for the sparse angle and the speed of sound:(1)$$\begin{array}{l} \tan\alpha =\sqrt{{\left(C/D\right)}^{2}-{\left[\left(D-U\right)/D\right]}^{2}}\\ C=D\sqrt{{\left(\tan\alpha \right)}^{2}+{\left[\left(D-U\right)/D\right]}^{2}} \end{array}$$

Thus, the velocity of sound can be obtained by measuring the sparsity angle, if the D and U of the shock wave are known, and the tangent of the rarefaction angle needs to be measured experimentally. However, due to the presence of elastic longitudinal waves, it is difficult to accurately determine the boundaries of the unloading zone using optical photographs, and the resulting measurement of sound velocity is not accurate. Then, in order to accurately measure the position of point B, a high-spatially resolved linear/area close-row velocimeter probe is required.

### 2.2. Line-Resolved Fiber Probe

All-fiber laser interferometry velocimetry, such as PDV (Photonic Doppler Velocimetry) [17], DISAR, etc., has been widely used in shock wave physics and other fields because of its simple structure and wider applicability of optical fiber transmission. However, to further improve the spatial resolution of this technology, it is necessary to design and optimize the front-end fiber probe structure.

Although the traditional single bare fiber probe can achieve spatial close distribution (bare fiber spacing ~150 μm), the depth of field is small (~1 mm) and needs to be close to the measured surface. Probes with focusing lenses have a large depth of field (tens of mm) and are suitable for single-point measurements, but their large transverse size (outer diameter > 1 mm) makes it difficult to achieve high spatial resolution. In this study, a spatially resolved probe was designed by means of single-mode fiber line array close-row and optical lens group coupling. The optical system is a large-range multi-point measurement system, and in order to improve the depth of field of the optical system, it is necessary to design a small numerical aperture of the object. At the same time, in order to ensure the consistency of the measurement of each point, the optical group of the object side telecentric three-lens with small numerical aperture is the best structural form. This is to take advantage of the long focal length and large depth of field of the long-beam three-light group and the small outer dimension, making it more suitable for smaller objects, as shown in the Figure 2. Compared with the previous fiber probes, the one-dimensional single-point measurement has been improved to the two-dimensional linear array multi-point measurement (25 measurement points). The spatial resolution can reach the diffraction limit (~20 μm), and more information can be obtained. Meanwhile a certain measurement depth of field (~4 mm) is guaranteed, and a longer speed-time history can be obtained.

The final probe specifications are shown in the Table 1.

In order to facilitate the measurement, the optical path system is inverted in the optical design software, the optical fiber close-row array is the object surface, and the measured target surface is the image plane, as shown in Figure 3 (The green, red, blue and yellow lines in Figure 3 represent the input light at different locations). A point light source with a numerical aperture of 0.14 and wavelengths of 1549 nm, 1552 nm, and 1554 nm is set at the location of each fiber. The point diagram of the system was analyzed under the sequence model as an index to evaluate the system design. The dot diagram at the focal and defocus locations is shown in the Figure 4, with an Airy spot radius of 5.5 μm.

The results of Table 2 show that the root mean square radius is less than 20 μm, and the spot interval of waist position is less than 20 μm, indicating that the optical design of the optical system realizes the ability of spatial resolution.

When the measured surface is in the out-of-focus position, the image square point column of the optical system is shown in the figure. As can be seen from the Figure 5, the center of the spot on the measured object is slightly shifted after defocus, but the overall shape is well maintained without large aberrations. where the negative amount of defocus corresponds to the probe of the analyte and the positive amount of defocus corresponds to the distance of the probe of the analyte. In addition, because the spot diameter is smaller than the spot spacing, all spots do not overlap after defocus, thus avoiding crosstalk between channels.

As shown in the Figure 6, the diameter of the probe is only 16 mm, and the spot spacing is less than 20 μm.

## 3. Experiments and Results

The experimental setup is mainly composed of a tantalum flyer, a step target of iron, and a linear-resolved fiber probe, as shown in the Figure 7. The impact loading experimental platform is the two-stage light gas gun platform, which uses explosives to drive compressed gas to accelerate the tantalum flyer, which hits the step target of iron at 2.54 km/s. The linear-resolved probe of 24 points is 4 mm away from the upper surface of the step, and the 18th measurement point is in the center of the step. In order to determine the position of the measuring point on the surface of the step, it is necessary to carefully move the probe along the step surface to observe the position of the spot under the high magnification microscope until the 18th measuring point is in the center of the step.

In the light gas gun experiment, the high-pressure gas expands in the launch tube of the two-stage light gas gun and drives the projectile to accelerate for about a few milliseconds. The acceleration process of the projectile is close to the quasi-isentropic shock-free drive. Adjusting the amount of gunpowder and the pressure of light gas can accurately control the target velocity of the flyer, so that the flyer is in a constant motion state when it reaches the outlet end of the launch tube and maintains an excellent plane flyer attitude. The difference between the physical state of the high-speed flyer at the time of target shooting and the physical state at atmospheric pressure is very small. The velocity of the flyer at the moment of impact can be accurately measured.

The “side sparse wave” method can also be used to measure the sound velocity after the shock wave of a metal. After the high-speed flyer hits the target, the sample is in the high-pressure state of the shock wave, but the side positions of the flyer, the target plate and the sample to be measured are all in the zero-pressure state, and the sparse wave from the side part of the experimental device will interfere with the propagation and post-wave state of the plane shock wave. The wave head of the lateral sparse wave propagates at the speed of sound c with respect to the material after the shock wave.

This is because the state does not change with time under the uniaxial strain load generated by the plane impact compression. The close-knit line/area laser velocimetry area covers the entire metal target surface, so that the response of the material’s geometry to shock loads can be obtained. Since sparse waves propagate inward from the edge of the target material, a spatially resolved velocimetry probe captures the influence boundary of the edge sparse wave (i.e., the location of point M or N in the diagram). Then, the sparse angle α of the side is measured, and the sound velocity after the shock wave of the metal material is reversed by Equation (1). When the impact pressure is high, the metal material melts by impact, and the measured sound velocity is the volume wave sound velocity, which provides experimental data for the equation of the state of matter of the material. A schematic diagram of the experimental setup is shown in Figure 8.

In order to reduce the backlash of the detection light, a glue with a matching refractive index should be applied between the end face of the fiber array and the lens. In addition, all lens surfaces should be coated with a highly anti-reflection dielectric film to reduce the interference between the return light on the end face and the signal light.

Iron samples are known to have a step height of 300 μm and a step diameter of 1000 μm. The eccentricity of the linear fiber optic probe is directly opposite the surface of the step, and the center of the step corresponds to the 18th measurement point of the probe. As can be seen from the linear velocity profile of Figure 9, the sparse area is preceded by the 8th measurement point, so the width of the sparse area on the step surface is 500 − 10 × 20 = 300 μm. Therefore AA_1_ = MM_1_ = 300 μm, so the α of the sparsity angle is about 45°.

It is known that *C*_0_ is 4.63 km/s, *λ* is 1.33, and *u* is half of the free surface velocity, i.e., 1.5 km/s. The velocity of the shock wave can be obtained from *D* = *C*_0_ + *λu*, *D* = 4.63 + 1.33 × 1.5 = 6.625 km/s. The shock wave pressure can be changed from *P* = *ρ*_0_*Du* = 7.85 × 6.625 × 1.5 ≈ 78 GPa.

## 4. Discussion

It is known that there are two ways to measure high-pressure sound velocity, one is the catch-up method and the other is the side-release method. The catch-up method has been the main method of measuring the sound velocity, and one of the optical analysis techniques is to observe the drop in optical radiation energy to determine the position and time of the shock wave front. Multi-step samples are often used, which are larger in size and have a limited number of platforms available. The other is the line-VISAR technique, which determines the sound velocity by looking at the velocity profile of particles at the interface through a window. This requires the window material to remain transparent during impact loading, so the impact pressure is limited (usually not higher than 200 GPa).

In this paper, we use the attenuation shock wave, which is a kind of triangular wave, and the shock wave array is in the same state at room temperature and pressure. With the decrease of the shock wave intensity, the post-wave state still satisfies the Hugoniot relation. That is to say, the post-wave state is to go from high to low along the Hugoniot line. The side-release wave introduced in the experiment always acts with the shock wave at the shock wave front immediately after the shock wave, so the state corresponding to the sound velocity is strictly on the Hugoniot line, and it changes continuously. By using this continuous pressure change, it is expected that the measurement of the complete equation of matter can be achieved with a small number of experiments through the continuous measurement of space by high line-spatially resolved DISAR.

The side-release method has not been well developed since it was proposed, and it mainly relies on cameras to obtain high-spatially resolved shock wave wavefronts. However, the camera can only acquire a few transient images and cannot obtain a continuous velocity-time history. With the development of laser interferometry such as DISAR, combined with high-spatially resolved fiber optic probes, the technique of side-release measurement of sound velocity can be used again.

This paper’s aim is to measure the sound velocity at a volume wave along the main Hugoniot. Under the fluid approximation model, the impact compression system has only one degree of freedom in the p-V-T state space. As long as any one of the state quantities in the (p, V, T) variables is determined, the rest of the state quantities are completely determined. The shock wave pressure measured in this experiment is 78 GPa, then the impact temperature T is 1420 K (calculated by the equation of state model). The sound velocity is 8.4 km/s, which is about 3% different from the results in the literature [18,19].

The object of measurement is high-purity polycrystalline iron, which is the main element of the earth’s core. The technology studied in this paper can be applied to any solid material, as long as it can reflect the laser light. This method can be used to measure high-pressure sound velocity of any solid material. And with the advantage of high spatial resolution, the sample size can be reduced to 100 μm, which is more suitable for the three-stage light gas gun, magnetic drive loading, and laser loading platforms. At the same time, there is no need to add a window, which avoids the bottleneck problem of opaque window material under high pressure. Of course, micro-scale step samples also pose a great difficulty for precision machining.

Li Mu combined the high-speed camera on the VISAR space optical path and used the side-release method to obtain the high-pressure continuous sound velocity of quartz [13]. But they only measured transparent materials, not metal materials like iron, because they had to record with a camera. It is one of the important ways to obtain geophysical knowledge at the interface between the inner and outer cores by using shock wave physics to study the melting laws of iron and ferroalloys under high pressure in the earth’s core. The technical approach studied in this paper provides the possibility of such research.

## 5. Conclusions

The study of ultra-high pressure sound velocity of iron is of great significance for the study of the composition and properties of matter in the high-temperature and high-pressure environment of the earth’s core. In this paper, a spatially resolved optical system probe is used to achieve a linear velocity measurement with a spatial resolution of better than 20 μm. On the two-stage light air cannon loading platform, the high-pressure sound velocity of pure iron was measured by the side sparse method, and the sound velocity value was 8.4 km/s under the shock wave pressure of 78 Gpa.

The high-spatially resolved linear velocity measurement DISAR technology is based on the original high-time-resolution displacement measurement, which extends the original single-point measurement to a one-dimensional linear velocity measurement and has been initially applied in high-pressure sound velocity measurement. It provides a new technology for the future application of high spatial resolution measurement.

## Figures and Tables

**Figure 1 micromachines-15-01165-f001:**
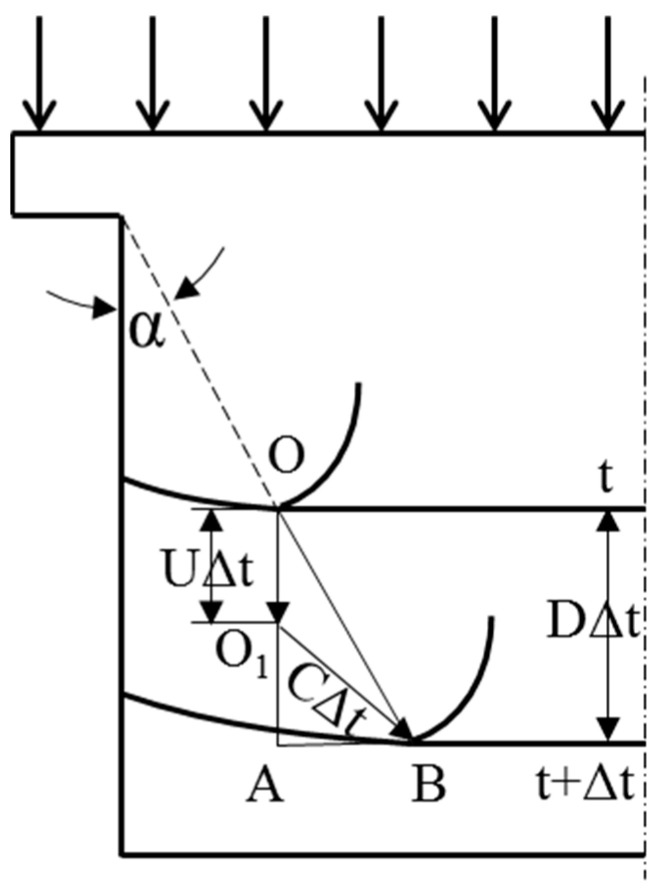
Schematic diagram of the effect of side sparse waves.

**Figure 2 micromachines-15-01165-f002:**
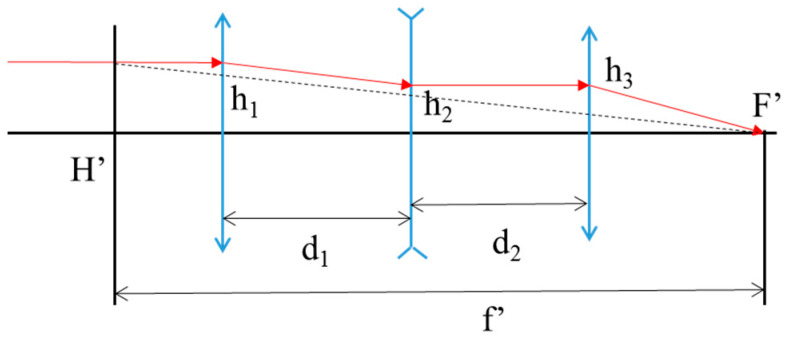
Long-range three-light group system.

**Figure 3 micromachines-15-01165-f003:**
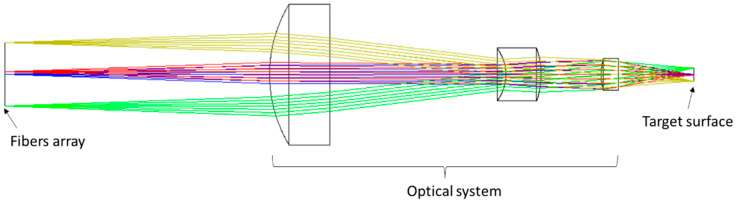
Diagram of a line-resolved optical system.

**Figure 4 micromachines-15-01165-f004:**
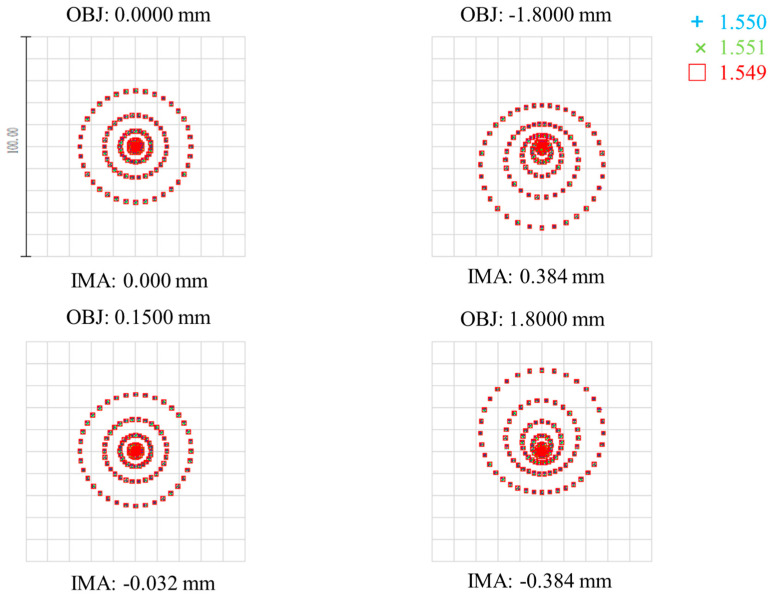
A spot diagram of each object position at the focal point.

**Figure 5 micromachines-15-01165-f005:**
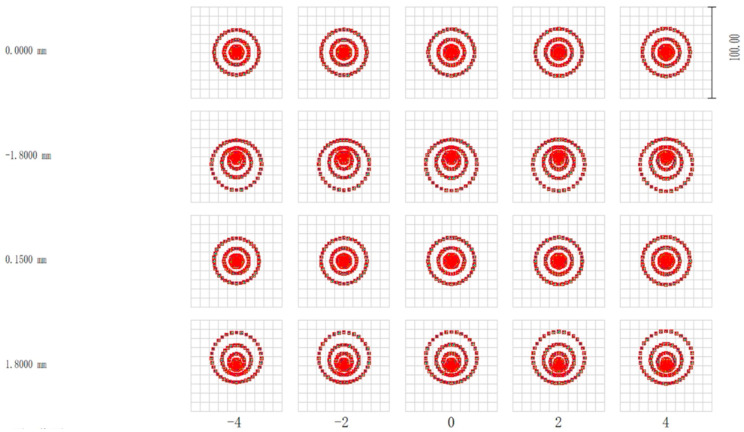
A spot diagram of each object position at the out-of-focus position.

**Figure 6 micromachines-15-01165-f006:**
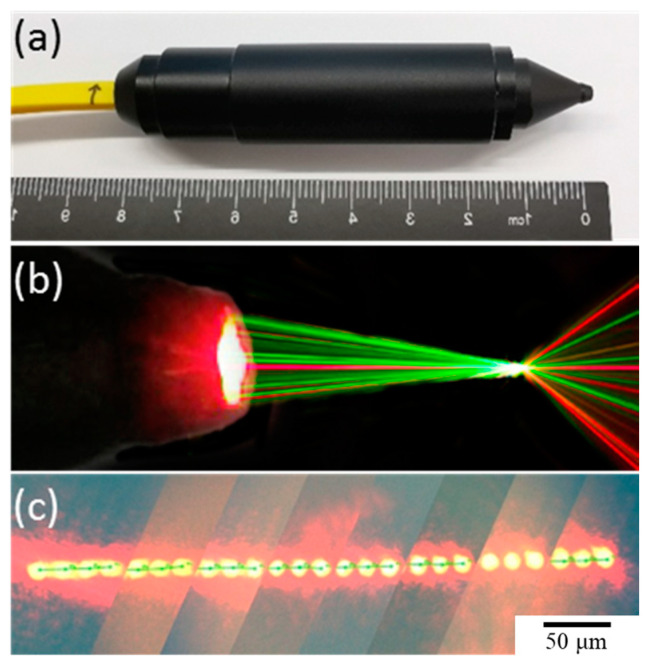
Photograph of line-resolved fiber probe. (**a**) the physical diagram, (**b**) light simulation diagram, and (**c**) light spots diagram of the line-resolved fiber probe.

**Figure 7 micromachines-15-01165-f007:**
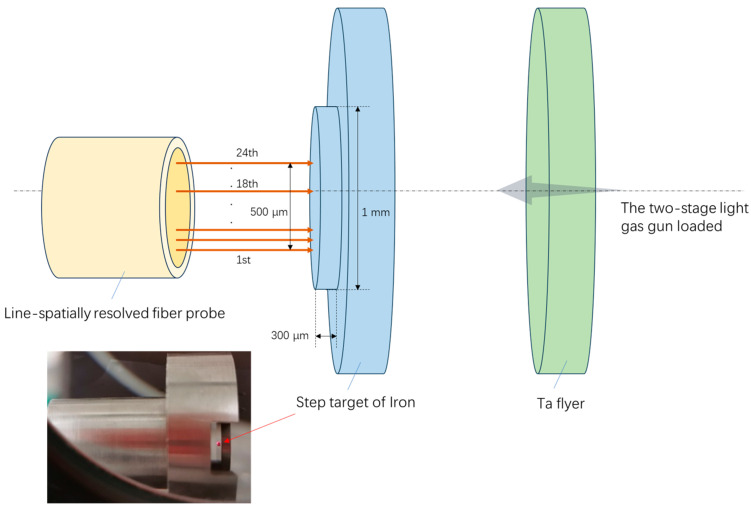
Schematic diagram of the relationship between the position of the experimental device and the measurement points of linear-spatially resolved fiber probe.

**Figure 8 micromachines-15-01165-f008:**
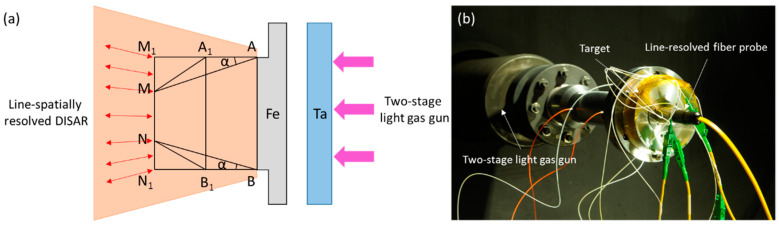
A schematic diagram of the experimental setup. (**a**) Linear-spatially resolved DISAR and side sparse areas. (**b**)Two-stage light gas gun.

**Figure 9 micromachines-15-01165-f009:**
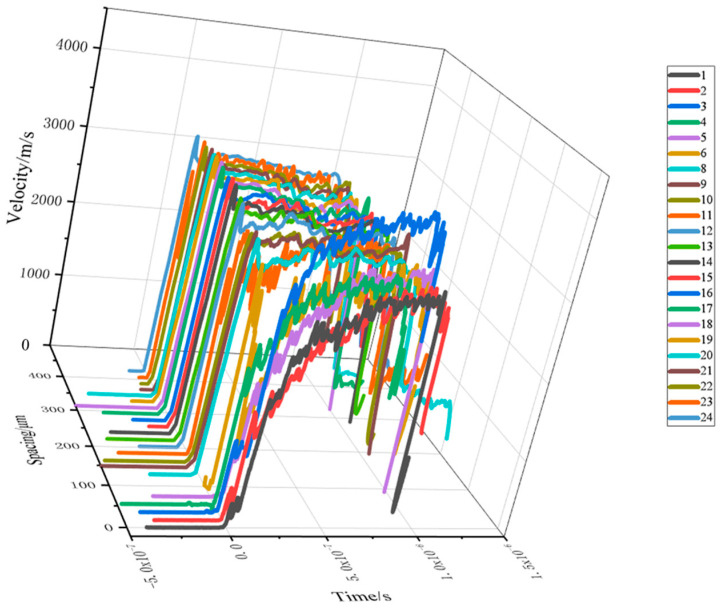
Linear velocity plot of a free-plane shockwave.

**Table 1 micromachines-15-01165-t001:** Parameters of spatial resolution DISAR probe.

Parameter	Specification
Fiber type	TEC fiber
Number of fibers	25
NA	0.07
Mode field diameter/μm	20
Wavelength/nm	1550
Spatial resolution/μm	20
Displacement of target/mm	4

**Table 2 micromachines-15-01165-t002:** Spot diagram parameters at three different positions of image plane.

No.	Waist Position			Waist Position − 1.8 mm		Waist Position + 1.8 mm	
	Coordinate of Fibers/μm	Spacing of Adjacent Spots/μm	RMS Diameter of Spots/μm	Spacing of Adjacent Spots/μm	RMS Diameter of Spots/μm	Spacing of Adjacent Spots/μm	RMS Diameter of Spots/μm
1	60	\	7.3	\	8.0	\	7.9
2	180	18	7.4	20	8.1	21	8.0
3	300	19	7.3	21	8.3	20	8.1
4	420	19	7.5	20	8.5	20	8.3
5	540	18	8.0	22	8.6	21	8.3
6	660	19	8.2	21	8.7	22	8.5
7	780	18	8.5	20	8.9	23	8.8
8	900	19	8.6	20	9.0	22	8.9
9	1020	19	8.8	21	9.0	22	8.9
10	1140	20	8.9	20	9.1	23	9.0
11	1260	19	8.9	21	9.2	22	9.2
12	1380	20	9.0	21	9.3	23	9.2
13	1500	20	9.0	21	9.3	23	9.3

## Data Availability

The original contributions presented in the study are included in the article, further inquiries can be directed to the corresponding authors.

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
