# Peer review of "The Side-Release Method Measures the High-Pressure Sound Velocity of Iron Using Line-Spatially Resolved DISAR"

_micromachines, 2024, doi:10.3390/mi15091165_

Round 1
Reviewer 1 Report
Comments and Suggestions for Authors
Review of “The Side-Release Method Measures the High-Pressure Sound Velocity of Iron Using Line-Spatially Resolved DISAR” by Chen et al. submitted to Micromachines
This paper reports technical developments for using the “side sparse wave” method in a two-stage light gas gun experimental platform to measure the sound velocity of iron at high-pressure conditions. To be honest, this paper is a bit too technical for me to judge, as I am not an expert in shock-wave experiments. So, I cannot say much about their technical details. And also, I’m not familiar with the journal Micromachines and whether this paper is suitable for this journal or not. I’ll leave these to other reviewers and the editor to judge. My major concern is in the discussion part of the paper, which is basically a summary of the developments that the authors have done. However, in order to test the feasibility or advantage of the technique as claimed by the authors, I think the sound velocity data should be compared with previous studies using other techniques. I’m pretty sure that there are tons of high-pressure sound velocity data available for iron in the literature, measured by both dynamic-compression experiments (e.g., shock-wave experiments) and static-compression experiments (e.g., diamond-anvil cell experiments), but such a comparison with previous results is totally missing in this paper, significantly reducing its credibility. It seems like that the authors only get one data point for the sound velocity of iron at 78 GPa and the temperature is not reported. To get the equation of state, one data point is definitely not enough.
Other issues:
Line 20. I’m not quite understand what the “mainland nuclear power” means here.
Line 26. The sentence “experimental method of ultra-high-pressure sonic velocity” does not make sense and is also similar to the next statement, thus it should be deleted.
Line 31. “sphere structure” should be “spherical structure”
Line 33. “Initial Reference Earth Model” should be “Preliminary Reference Earth Model (PREM)”
Line 41-42. I don’t understand this statement. What do the authors mean by saying the “mechanism of its sound velocity”? What kind of mechanism does sound velocity have? To me, sound velocity is a property related to the material itself, and it’s a bulk property of the material, so I don’t understand how the surface would affect its mechanism.
Line 58. Should be “is in accordance with”.
Line 59-60. I don’t think this statement is correct. There are numerous literatures reporting the sound velocity of iron at high pressures using different techniques.
Line 194. Which figure do the authors refer to?
Comments on the Quality of English LanguageThe English is fine, just need some minor editing of grammatical errors and typos.
Reviewer 2 Report
Comments and Suggestions for Authors
The manuscript, titled "The Side-Release Method Measures the High-Pressure Sound Velocity of Iron Using Line-Spatially Resolved DISAR", by Chen et al. introduces a measurement method for high-pressure sound velocity. For the publication of this manuscript, it needs some necessary revisions to give clearer and complete explanation. My comments and suggestions are listed in the following.
(1) Clearly specify the novelty of the study in the introduction section. I understand that there are same reported results on this subject but what is different here?
(2) The authors can discuss a bit about the potential application of the present method in the introduction section.
(3) Page 3, line 132, Page 3, line 132, the second is the research on multi-channel laser Doppler velocimetry technology based on wavelength division time division multiplexing technology. Please check for any errors: is it wavelength division multiplexing or time division multiplexing?
(4) To achieve precise measurements, a high-spatially resolved linear/area close-row velocimeter probe is required. The authors should provide a detailed explanation of how the optical fiber probe has been optimized. What are the innovative aspects? What improvements in performance metrics have been achieved compared to previous versions? In particular, “Long-range three-light group system”, Please provide a detailed explanation.
(5) In the experimental section, the experimental setup should be further introduced, including the components and functions of the experimental platform, the experimental conditions, the experimental process, and how to control the factors affecting the experiment, all of which need to be explained.
(6) I suggest rewriting the discussion section. The discussion should analyze and discuss the measurement methods, measurement subjects, and measurement results in the current work, rather than simply listing the results.
Comments on the Quality of English Language​
Round 2
Reviewer 2 Report
Comments and Suggestions for Authors
The quality of the manuscript has significantly improved. It is recommended to further check for spelling and grammar to enhance readability for most scholars.